# Imaging Mitochondrial Functions: From Fluorescent Dyes to Genetically-Encoded Sensors

**DOI:** 10.3390/genes11020125

**Published:** 2020-01-23

**Authors:** Elif Begüm Gökerküçük, Marc Tramier, Giulia Bertolin

**Affiliations:** Univ Rennes, CNRS, IGDR [Institut de génétique et développement de Rennes] UMR 6290, F-35000 Rennes, France

**Keywords:** mitochondria, Ca^2+^ signalling, mitochondrial dynamics, mitophagy, fluorescence microscopy, chemical dyes, genetically-encoded sensors, super-resolution microscopy

## Abstract

Mitochondria are multifunctional organelles that are crucial to cell homeostasis. They constitute the major site of energy production for the cell, they are key players in signalling pathways using secondary messengers such as calcium, and they are involved in cell death and redox balance paradigms. Mitochondria quickly adapt their dynamics and biogenesis rates to meet the varying energy demands of the cells, both in normal and in pathological conditions. Therefore, understanding simultaneous changes in mitochondrial functions is crucial in developing mitochondria-based therapy options for complex pathological conditions such as cancer, neurological disorders, and metabolic syndromes. To this end, fluorescence microscopy coupled to live imaging represents a promising strategy to track these changes in real time. In this review, we will first describe the commonly available tools to follow three key mitochondrial functions using fluorescence microscopy: Calcium signalling, mitochondrial dynamics, and mitophagy. Then, we will focus on how the development of genetically-encoded fluorescent sensors became a milestone for the understanding of these mitochondrial functions. In particular, we will show how these tools allowed researchers to address several biochemical activities in living cells, and with high spatiotemporal resolution. With the ultimate goal of tracking multiple mitochondrial functions simultaneously, we will conclude by presenting future perspectives for the development of novel genetically-encoded fluorescent biosensors.

## 1. Introduction

Mitochondria are essentially known for their role in adenosine triphosphate (ATP) synthesis and contribution to cellular metabolism, yet these organelles regulate a multitude of cellular functions. These functions include Ca^2+^ buffering, cellular signalling, reactive oxygen species (ROS) production, and apoptotic cell death [1,2,3,4]. Mitochondria are also the major sites where the key steps of heme biosynthesis, ketone bodies generation, and hormone synthesis are performed [5,6,7]. In response to different physiological or pathological cues, mitochondria form highly dynamic networks to meet the metabolic demands of the cellular sub-domains [8]. Similarly, at the tissue level, mitochondria within a group of specialized cells can show a trend towards a certain mitochondrial network architecture. This parameter was shown to be of importance with respect to the bioenergetic efficiency of the tissue [9]. Furthermore, mitochondria have their own genome containing 37 genes, and 13 of them encode the core protein subunits of the oxidative phosphorylation system [10]. 

This functional and morphological diversity of mitochondria is important for the maintenance of cellular homeostasis. Therefore, it is not surprising that mitochondrial defects play a major role in many pathologies, including neurodegeneration, aging, cancer, diabetes, obesity, and cardiomyopathies [11]. In order to better understand not only the fundamental mitochondrial biology, but also the mitochondrial pathophysiology, imaging techniques with chemical dyes and genetically-encoded sensors provide the opportunity of gathering detailed insights of complex mitochondrial functions. In this review, we will be addressing three key mitochondrial functions: Calcium signalling, mitochondrial dynamics, and mitophagy. After a brief description of the molecular mechanisms underlying these functions, we will discuss the selection of probes that are available to study these complex mitochondrial functions.

## 2. Mitochondria and Ca^2+^


Understanding the role of Ca^2+^ in physiology dates back to the observations by Sydney Ringer in 1883, where he showed that the addition of Ca^2+^ to the perfusion buffer of isolated hearts triggered their contraction [12]. Almost 80 years after this finding, isolated mitochondria have been shown to accumulate high amounts of Ca^2+^ using respiratory energy (under the form of ATP) [13,14]. With the continuous development of probes and methods for Ca^2+^ imaging over the years, mitochondria are now known to be important regulators of cellular Ca^2+^ levels and Ca^2+^ signalling. This coordination was shown to be required for cellular metabolism, survival, and cell-type specific functions. In terms of Ca^2+^ buffering and signalling, mitochondrial activity depends on their close communication with endoplasmic reticulum (ER) (sarcoplasmic reticulum in muscle cells) [15]. 

### 2.1. Mitochondrial Ca^2+^ Transport

Upon stimulation with agents, increasing Ca^2+^ concentration ([Ca^2+^]), mitochondria have the potential to accumulate from 10 to 20-fold more Ca^2+^ than what is stored in the cytosol [16]. The driving force behind the accumulation of positively charged Ca^2+^ ions in the mitochondrial matrix is the membrane potential difference (ΔΨ) generated by the respiratory chain. To be able to reach the mitochondrial matrix, Ca^2+^ ions have to cross both the outer mitochondrial membrane (OMM) and the inner mitochondrial membrane (IMM) (Figure 1). Voltage-dependent anion-selective channel proteins (VDACs), located on the OMM, are the first barriers regulating Ca^2+^ uptake, and their high expression in cultured cells was shown to ensure permeability [17]. Once the Ca^2+^ ions are in the intermembrane space (IMS), they reach the matrix via the mitochondrial Ca^2+^ uniporter (MCU) complex [18,19]. The MCU is a macromolecular complex containing pore-forming subunits located on the IMM, and several regulatory proteins protruding into the IMS [20,21,22]. While MCU-dependent Ca^2+^ uptake is highly selective, this complex has a low dissociation constant (*K_D_*) for Ca^2+^ [23]. The tight control of MCU opening is controlled by the Ca^2+^-sensing regulatory proteins mitochondrial Ca^2+^ uptake 1 (MICU1) and mitochondrial Ca^2+^ uptake 2 (MICU2), which contain EF-hand motifs directly binding to Ca^2+^ ions [24,25]. MICU 1/2 serve as gatekeepers of the MCU complex by reacting with the cytoplasmic Ca^2+^ concentration ([Ca^2+^]_c_). When the [Ca^2+^]_c_ is low, MICU 1/2 interact with the central MCU unit to keep the gate closed and prevent mitochondrial Ca^2+^ uptake [26,27,28]. Conversely, the release of Ca^2+^ from ER results in an increase of local [Ca^2+^]_c_ at the ER-mitochondria interface. Since Ca^2+^ binds the EF hand domains of MICU 1/2 when the [Ca^2+^]_c_ is high, this binding triggers a conformational change of MICU 1/2 that results in the opening of the MCU [28].

While the MCU complex regulates Ca^2+^ influx into the mitochondrial matrix, efflux of Ca^2+^ is regulated by the mitochondrial Na^+^/Ca^2+^ exchanger (mNCX) and the mitochondrial H^+^/Ca^2+^ exchanger (mHCX) [29,30,31]. Since the action of mNCX might trigger excess Na^+^ accumulation in the matrix, this is then calibrated by the Na^+^/H^+^ exchanger, which extrudes Na^+^ in the IMS [32]. The mitochondrial permeability transient pore (mPTP) has also been shown to have a role in Ca^2+^ efflux in cardiac cells to regulate the cellular metabolism according to the myocardial workload [33]. Interestingly, transient mPTP opening in myocytes were also proposed to act as “reset mechanisms” for mitochondria, counteracting a Ca^2+^ overload or dissipating high ROS damage [34]. Last, the leucine zipper-EF-hand-containing transmembrane protein 1 (LETM1) has been proposed as a mitochondrial Ca^2+^/H^+^ antiporter in a genome-wide RNAi screen study [35]. However, these findings are still debated, and a clear role of mPTP and LETM1 in Ca^2+^ efflux has not been established yet [36,37,38].

### 2.2. Role of Mitochondria in Ca^2+^ Homeostasis

Changes in mitochondrial Ca^2+^ concentration ([Ca^2+^]_mito_) have been shown to regulate many cellular processes such as ATP production by oxidative phosphorylation [39,40,41], apoptosis [42,43,44], autophagy [45,46], and organelle crosstalk between the mitochondria, and ER [47]. The ability of mitochondria to capture Ca^2+^ molecules released at the Ca^2+^-signalling microdomains—formed between mitochondria and plasma membrane or at the ER-mitochondria junctions—can modulate the activity of Ca^2+^ channels by modifying the local [Ca^2+^]_c_ in these microdomains [48,49]. Mitochondria can also contribute to the accumulation of Ca^2+^ ions by forming a “mitochondrial belt”. This structure, described in specific cellular types such as neuronal cells [50,51,52,53,54] or in pancreatic acinar cells [55,56,57,58], acts as a physical barrier, preventing Ca^2+^ diffusion away from defined subcellular domains. 

Cellular metabolism and survival can also be affected by mitochondrial Ca^2+^ uptake. On the one hand, the localization of mitochondria to the sites of high [Ca^2+^]_c_, followed by the MCU-mediated [Ca^2+^]_mito_ increase, activates the Ca^2+^-sensitive dehydrogenases of the Krebs cycle and the ATP synthase, ultimately leading to ATP production [59,60]. On the other hand, excessive Ca^2+^ overload in the mitochondrial matrix is known be associated with apoptosis, due to the sustained activation of mPTP and the release of apoptosome components such as cytochrome *c* into the cytosol [61]. 

Overall, the interplay between cellular Ca^2+^ and mitochondrial Ca^2+^ plays an important role in energy production and in a variety of signalling processes.

### 2.3. Probes to Measure Mitochondrial Ca^2+^

Of all the probes generated to monitor mitochondrial functions, the ones dedicated to track Ca^2+^ ions and to measure their concentration are the most abundant (Figure 1). Probes monitoring intracellular Ca^2+^ or sensors measuring changes in Ca^2+^ levels within mitochondria and other organelles have thoroughly helped researchers to elucidate the broad spectrum of functions that are regulated by Ca^2+^ ions. 

The choice of the most suitable tool(s) from the palette of Ca^2+^ probes currently available depends on the type of assay and the desired readout. To this end, several elements must be taken into account. First, the type of probe to be used is a crucial parameter, as the choice must be made between chemical vs. genetically-encoded Ca^2+^ indicators. Second, probe(s) to monitor Ca^2+^ in living cells must be selected according on the cell type of interest and on the overall duration of the experiment to perform. This parameter is of key importance, as it should be carefully chosen to optimize the sensitivity and spatiotemporal accuracy of the Ca^2+^ probe. Last, the Ca^2+^ affinity and spectral properties of the probe should also be taken into account [62,63]. These elements are discussed in the sections below.

#### 2.3.1. Chemically-Engineered Ca^2+^ Indicators

Initial attempts to measure cytosolic Ca^2+^ have been performed with chemically-engineered fluorophores that change their fluorescence properties upon Ca^2+^ binding. These Ca^2+^ indicators have varying affinities for Ca^2+^, along with different spectral properties. However, most of them are acetoxymethyl ester (AM)-based compounds [64]. AM-based Ca^2+^ indicators are hydrophobic and primarily diffuse into the cytoplasm, where they are hydrolyzed by the cellular esterases and become trapped inside the cells [64,65]. While this strategy overcomes the problem of the complexity of microinjections, it cannot be used to specifically target organelles as mitochondria.

The spectral changes that can be observed in Ca^2+^ indicators upon Ca^2+^ binding are: (i) Change in fluorescence intensity without shifts in the excitation/emission wavelengths, as in single wavelength indicators, or (ii) shift in excitation and/or emission spectra, as in ratiometric indicators. Classical examples of Ca^2+^ indicators can be Fura-2 and Indo-1, which were originally developed by Tsien et al. in the 1980s [66]. Here, Ca^2+^ binding can either cause a shift in the excitation or emission wavelength of Fura-2 and Indo-1, respectively. In the case of other indicators such as Fluo-3 and Calcium-Green, Ca^2+^ binding increases the emitted fluorescence. While these indicators are mainly used to monitor intracellular Ca^2+^, Rhod-2 of the rhodamine-based family of indicators represents an exception in terms of mitochondrial Ca^2+^ monitoring. The advantage of Rhod-2 is its net positive charge (both the AM and the hydrolysed forms), which promotes the accumulation of Rhod-2 into the mitochondrial matrix and a fluorescence increase upon Ca^2+^ binding [64]. However, Rhod-2 diffuses out of mitochondria soon after its entry into these organelles, causing inaccuracy during long time-course experiments. 

Today, a broad spectrum of chemically-engineered Ca^2+^ indicators offers solutions for many biological questions related to Ca^2+^. Some of these indicators, like Rhod-2 and Fura-2, are not only used to report qualitative Ca^2+^ level changes, but they are also useful to measure the exact [Ca^2+^]_mito_ [67,68,69]. However, and in addition to accurate targeting problems, the toxicity caused by the de-esterification reaction during a prolonged excitation is an important disadvantage related to the use of these probes [65,70,71].

#### 2.3.2. Genetically-Encoded Ca^2+^ Indicators

Our knowledge of Ca^2+^ signalling has been accelerated with the creation of genetically-encoded Ca^2+^ indicators (GECIs). These probes offer convenient solutions for the problems associated with chemical Ca^2+^ indicators. GECIs possess either bioluminescent (based on aequorin) or fluorescent (based on green fluorescent protein (GFP) and its derivatives) proteins to report changes in Ca^2+^ signalling [72,73,74,75,76]. One of the biggest advantages that GECIs offer is that they can be targeted to desired organelles or cytoplasmic domains when they are fused with specific targeting signal peptides. Their expression can also be spatiotemporally controlled via tissue-specific or inducible promoters. Moreover, as GECIs are genetically-encoded, they offer a better substrate specificity and less variance in probe uptake, as compared to chemical Ca^2+^ indicators [77]. However, special attention must be given when choosing a specific GECI to use. GECIs are larger in size, as compared to chemical Ca^2+^ indicators, and their expression might potentially alter mitochondrial morphology or functions [65]. Similar to chemical Ca^2+^ indicators, several GECIs can be employed to measure the exact [Ca^2+^]_mito_ [78,79,80].

Bioluminescence-based GECIs are derived from the Ca^2+^-sensitive photoprotein Aequorin (Aeq) isolated from the *Aequorea victoria* jellyfish [81]. Aeq has EF-hand motifs for Ca^2+^ binding and a hydrophobic core that can bind to an external cofactor like coelenterazine. Upon Ca^2+^ binding, Aeq undergoes an irreversible reaction in the presence of coelenterazine, and produces a photon of light [82]. The speed of emitted light can then be used to determine [Ca^2+^] [83]. A bioluminescence-based GECI named mtAEQ, which was composed of the mitochondrial targeting sequence of the cytochrome *c* oxidase polypeptide VIII (COX8) and fused with HA1-tagged native Aeq, was the first organelle-targeted GECI developed [84]. In fact, the first, direct evidence of mitochondrial Ca^2+^ accumulation in living cells upon stimulation has been made possible with the use of the mtAEQ probe. While bioluminescence-based GECIs have led the way to organelle-specific targeting, they have the disadvantage of being dim, as compared to fluorescence-based GECIs [85]. In addition, long-term imaging with these GECIs is not possible, due to the consumption of Aeq during the course of the reaction [86].

Fluorescent GECIs are comprised of a Ca^2+^-sensing polypeptide and can trigger a change in the fluorescence properties of the fused fluorescence protein. Fluorescent GECIs can be grouped into two classes: Single-fluorophore and Förster resonance energy transfer (FRET)-based GECIs. While single fluorophore GECIs exhibit changes in the fluorescence intensity or wavelength, FRET-based GECIs benefit from the energy transfer potential of the two fluorophores with partially overlapping excitation/emission spectra. 

One of the first examples of single fluorophore GECIs is the Camgaroo family indicator. These indicators have a calmodulin-Ca^2+^ binding domain, and they undergo a shift in the absorbance peak of the fluorescence protein upon Ca^2+^ binding [87]. Another example of single-fluorophore GECIs is the GCaMP. GCaMPs rely on the circular permutation of GFP: The N- and C-termini are fused, thereby creating two new N- and C-termini. These termini are then fused to the M13 domain of a myosin light chain kinase, and calmodulin-binding domain, respectively [88]. Ca^2+^ binding to the calmodulin moiety of GCaMP leads to changes in the chromophore environment, which results in an increased fluorescence intensity. Ever since their discovery, GCaMP GECIs have been extensively improved in terms of their spectral properties, Ca^2+^ affinity, brightness, and kinetics [89]. 

The very first examples of the FRET-based GECIs are cameleons, consisting of blue and green fluorescent proteins, acting as donor and acceptor FRET pairs. These fluorophores are connected with a calmodulin domain fused to a myosin light chain kinase M13 [90]. The principle behind the Ca^2+^-sensing function of the cameleons rely on the FRET phenomenon, which is a non-radiative energy transfer between a donor and an acceptor fluorophore. FRET can only occur when the donor-acceptor pair is in close proximity (<10 nm) and when the emission spectrum of the donor fluorophore overlaps with the excitation spectrum of the acceptor [91,92]. In the case of FRET-based GECIs, Ca^2+^ binding to the calmodulin domain changes the conformation of the probe, bringing the donor and acceptor pair into close proximity. This interaction allows the FRET reaction to occur, thus the changes in FRET efficiency can be directly used to estimate changes in [Ca^2+^] [93]. Similarly to GCaMPs, cameleons have been extensively improved since their first discovery. New versions of cameleons now show a decreased sensitivity to acidic pH, along with higher fluorescence intensities and decreased propensity to photobleaching [94]. 

In addition to bioluminescence and FRET-based GECIs, a new GECI family named GECO indicators, have been recently developed. They were engineered by performing directed evolution on GCaMP3 [95]. GECOs were shown to measure [Ca^2+^] in two different organelles or in different mitochondrial compartments simultaneously. Compared to FRET-based GECIs, GECO indicators have better signal-to-noise ratios, and they allow for multicolour imaging of Ca^2+^ ions in different organelles, or in different compartments of an organelle [96]. They are currently available with red, blue, and green intensiometric emissions [95].

As described in this section, the generous palette of chemically and genetically engineered Ca^2+^ indicators portrays the complexity of Ca^2+^ homeostasis within the cell. Thus far, being among the most fruitful sub-fields in mitochondrial research, elucidating the interplay between mitochondria and Ca^2+^ signalling will not only deepen our fundamental knowledge about the physiology of the cell, but it will also encourage the development of more sophisticated microscopy-based tools.

## 3. Mitochondrial Dynamics

The dynamic properties of mitochondria have been initially reported in the early 1900s, owing to the advancements in the field of bright-field microscopy [97]. Later, in the 1990s, the use of fluorescent dyes and proteins coupled to time-lapse microscopy revealed the first detailed insights about the changes in the mitochondrial morphology in living cells [98,99,100,101,102]. Mitochondria are very dynamic organelles, constantly undergoing fusion and fission events. These events are the interconnection of the organelles or the fragmentation of them into smaller units, respectively [103]. A wide series of reports over the last two decades demonstrated that changes in mitochondrial morphology have impacts on cellular metabolism, apoptosis, immunity, cell cycle, and mitochondrial quality control [104,105,106]. Furthermore, pathogenic mutations in genes coding for mitochondrial fusion and fission proteins have been associated with severe developmental defects and neuromuscular and central nervous system syndromes in mice and humans [107,108].

### 3.1. Molecular Players of the Mitochondrial Dynamics

Mitochondrial dynamics refers to the balance between mitochondrial fusion and fission events to regulate the size, shape, number, and distribution of mitochondria in cells, and this balance is constantly adjusted in response to physiological cues (Figure 2) [107]. On the one hand, mitochondrial fusion is the interconnection of individual mitochondria by joining their respective outer and the inner membranes, along with the sharing of intramitochondrial content as mitochondrial DNA (mtDNA) molecules. Mitochondrial fission, on the other hand, is the division of a mitochondrion into two or more mitochondrial units [109]. 

The balance between the opposing events of fusion and fission is not only required for proper mitochondrial functioning, but it is also necessary to adapt the mitochondrial network to the different metabolic states [105]. Mitochondrial fusion is required to regulate the respiratory chain activity, and maintains mtDNA integrity against the arousal of mutations or mitochondrial stress conditions [110,111]. In addition, pronounced fusion mechanisms during nutrient starvation were shown to protect the mitochondrial network from clearance through autophagy (mitophagy) [112]. Mitochondrial fission, however, is needed for the segregation and the consequent elimination of damaged mitochondrial parts during mitophagy, and for the inheritance of mtDNA to daughter cells during cell division [113,114]. 

The key molecular players of the mitochondrial fusion and fission events are GTPase proteins, belonging to a highly conserved Dynamin family [115]. Mitochondrial fusion or fission is achieved by the ability of these proteins to oligomerize and change conformation to trigger membrane modeling [116]. Mitochondrial fusion starts with the fusion of the OMM and in mammals, this is ensured by mitofusin 1 (MFN1) and mitofusin 2 (MFN2) [115]. The fusion event starts with the tethering of two mitochondria by homo- and hetero-oligomerization of MFNs [117]. Then, the conformational change of MFNs induced by the hydrolysis of GTP leads to the docking of the two adjacent membranes, while gradually increasing the membrane contact sites, and finally fusing the two OMMs [118,119,120]. Following OMM fusion, the dynamin-like GTPase optic atrophy 1 (OPA1) and specific IMM lipid components such as cardiolipin mediate IMM fusion [121]. OPA1 has sites for proteolytic cleavage that can be processed by IMM-bound metalloproteases OMA1 and YME1L, to generate shorter forms of OPA1 (S-OPA1) [122,123,124,125]. While the longer forms of OPA1 (L-OPA1) has been shown to be sufficient to drive fusion [126], S-OPA1 were suggested to couple IMM fusion to metabolism [127]. While there are different models describing IMM fusion, the interaction between L-OPA1 and cardiolipin was described to trigger the tethering of the two IMM, which is then followed by the membrane fusion upon OPA1-dependent GTP hydrolysis [121]. 

Mitochondrial fission is mediated by the dynamin-related protein 1 (DRP1), which is a large cytosolic GTPase protein recruited to mitochondria upon fission initiation [128]. Mitochondrial sites where DRP1 will be recruited are initially marked by ER tubules to be constricted [129]. It has recently been discovered that the replicating mtDNA is also found in the mitochondrial fission sites, in order to be able to allow the distribution of mtDNA to fragmented mitochondria [130]. Initiation of mitochondrial fission and mtDNA synthesis on the ER-mitochondria contact site are followed by the oligomerization of DRP1 on the OMM. DRP1 constricts mitochondria by forming ring-like structures around the organelles while hydrolyzing GTP [131,132]. Actin polymerization is also required at this step to facilitate DRP1 accumulation [133]. Moreover, adaptor proteins such as MID49, MID51, and the mitochondrial fission factor (MFF), serve to recruit DRP1 by acting as receptors or recruitment factors for DRP1 [134,135,136]. Lastly, Dynamin 2 (DNM2) recruitment to the DRP1-mediated mitochondrial constriction sites finalizes the mitochondrial membrane scission [137]. As opposed to the well-described mechanisms of OMM constriction and scission, fission machinery, specifically dedicated to IMM, is still being investigated. Some studies claimed that IMM division is Ca^2+^-dependent as this is supported by the observation of MCU loss leading to mitochondrial elongation [138,139,140]. The IMM protein MTP18 was also proposed to have a role in IMM fission, since its depletion results in mitochondrial hyperfusion and its overexpression causes mitochondrial fragmentation [141].

### 3.2. Probes and Methods to Study Mitochondrial Dynamics

Early studies focusing on mitochondria and their structural features generally rely on electron microscopy data. These approaches were the first ones to reveal that mitochondria have two membranes: A smooth outer membrane, and a convoluted inner membrane organized in cristae and projected into the mitochondrial matrix [142]. While electron microscopy techniques paved the way to our understanding of the mitochondrial ultrastructure, they cannot be employed to monitor fast changes in mitochondrial dynamics in living cells. To this end, advances in fluorescent microscopy techniques and tools were needed to study mitochondrial dynamics in real time and in living cells (Figure 2).

#### 3.2.1. Mitochondria-Specific Fluorescent Dyes and Proteins

Mitochondria can be labeled in living cells by using several fluorescent dyes that are mainly based on mitochondrial ΔΨ [143]. Examples of such cationic, ΔΨ-sensitive fluorescent probes are Rhodamine 123 (R123), tetramethylrhodamine ethyl ester (TMRE), tetramethylrhodamine methyl ester (TMRM), and 5,5,6,6’-tetrachloro-1,1’,3,3’ tetraethylbenzimidazoylcarbocyanine iodide (JC-1). In order to label mitochondria with these fluorophores, mitochondria should be functional to generate membrane potential. While these dyes have been used to monitor ΔΨ in several cell types, they are not the ideal probes to study dynamic changes in mitochondrial morphology since they leak from mitochondria upon a loss in mitochondrial ΔΨ [144]. The exception among them is the MitoTracker Green, due to its ability to bind to the free thiol groups in the mitochondria. This allows the dye to maintain its intramitochondrial localization even after a loss in mitochondrial ΔΨ [145,146]. The other available MitoTracker dyes are MitoTracker Red, Orange, and Deep Red. However, their sensitivity to mitochondrial ΔΨ is debated and may differ depending on the experimental setup [145,147,148,149]. Recently, alternative fluorescent probes for mitochondria based on naphthalimide (NPA-TPP) [150] or boron-dipyrromethene (BODIPY) [151] show better photostability, higher brightness, and lower cytotoxicity, thus enabling researchers to monitor changes in mitochondrial morphology in long time-course experiments [150,151].

Another approach to label mitochondria is to use fluorescent proteins targeted specifically to these organelles. Here, the idea is to fuse a fluorescent protein gene with a mitochondrial targeting sequence (MTS). Depending on the desired readout, different types of MTSs can be used to target fluorescence proteins to different mitochondrial sub-compartments. Moreover, two or more of the same MTS can be put in a series to facilitate the targeting of the fluorescent protein [152]. Once the construct is ready, it is essential to verify that the genetically-encoded fusion product is targeted properly to the desired mitochondrial sub-compartment. To do so, mitochondria can be imaged in the presence of dyes (such as MitoTrackers) to verify that the genetically-encoded fluorescent protein is not causing any change in the mitochondrial morphology. The validated construct can then be used to acquire time-lapse fluorescence videos to monitor changes in mitochondrial morphology. Acquired data can then be analyzed with conventional software programs designed to calculate size, shape, number, and the interconnectivity rate of mitochondrial objects.

#### 3.2.2. Imaging Methodologies with Genetically-Encoded Fluorescent Proteins

Due to the complex nature of mitochondrial dynamics, analyzing mitochondrial morphological changes alone is not sufficient to quantify fusion and fission events [153]. To this end, the use of mitochondria-targeted photoactivable and photoconvertible forms of fluorescent proteins has enabled researchers to differentiate between these events. Moreover, fluorescence recovery after photobleaching (FRAP) or fluorescence loss in photobleaching (FLIP) methods with regular mitochondrially-targeted fluorescence proteins provided the means to study diffusion properties within mitochondria.

A photoactivable GFP, named PAGFP, increases its fluorescence intensity 100-times after photoactivation [154]. By analyzing green-only pixels upon photoactivation, mitochondrial dynamics can be monitored in living cells expressing mitochondria-targeted photoactivable GFP (PAGFPmt) [154,155]. If the photoconverted PAGFPmt molecules are transferred to unlabeled mitochondrial units, this is indicative of fusion events, whereas a fission event can be followed if the PAGFPmt signal continuity is lost [153]. Photoactivable probes can also be used in combination with regular mitochondrial fluorescent proteins to distinguish non-fusing mitochondria. In order to do so, mitochondria that are co-expressing PAGFPmt and a mitochondrially-targeted DsRed fluorescent protein (mtDsRed) can be photoactivated for PAGFPmt, while bleaching mtDsRed in a region of interest by using 2-photon laser stimulation. In the resulting mitochondrial population, non-fusing units can only be identified with PAGFPmt signal in the region of interest. Fusing units, however, would show overlapping signals of photoactivated PAGFPmt and mtDsRed, if they interconnect with the surrounding non-photobleached and non-photoconverted mitochondria residing outside the region of interest [153].

Similar to photoactivation approaches, the photoconversion of genetically-encoded fluorescent proteins can be used to evaluate the interconnectivity of mitochondria. Photoconvertible fluorescent proteins employ an irreversible change in the fluorescence excitation and emission spectra upon an excitation at a specific wavelength [156]. A commonly employed example of this type of probe is Dendra2, which is a fluorescent protein switching from green to red when excited with a 405 nm laser [157]. In this context, mitochondrially-targeted Dendra2 (mitoDendra2) can be photoconverted to follow the diffusion of red molecules over green ones, thereby monitoring the continuity of the mitochondrial network [158]. Thus far, mitoDendra2 has been used to analyze mitochondrial interconnectivity in cultured cells and in several model organisms, including *C. elegans* [159], *Drosophila* [160,161], and mice [158].

Apart from photoactivable and photoconvertible fluorescent proteins, FRAP or FLIP methods can be utilized with regular mitochondrially-targeted fluorescent proteins to study mitochondrial dynamics. In FRAP experiments, a region of interest is photobleached and the fluorescence recovery in the bleached area over time is followed during time-lapse acquisitions [162]. As the recovery is due to the diffusion of fluorescent proteins adjacent to the bleached areas, this can be used as an indication of the interconnectivity of mitochondria [163]. In FLIP experiments, repeated photobleaching of a region of interest is performed while imaging the rest of the cell. This allows for the monitoring of the gradual loss in the fluorescence signal due to the movement of fluorescent proteins [162]. If the mitochondrial network is interconnected with the mitochondrial units in the bleached area, a decrease in the fluorescence signal in the rest of the cell is observed [162].

#### 3.2.3. Super-Resolution Microscopy for Mitochondrial Ultrastructure

Considering the size of a typical mitochondrion being close to the resolution limit of a conventional fluorescence microscope, super resolution microscopy techniques offer the opportunity to investigate the mitochondrial ultrastructure with their superior resolution. Some of the commonly available super resolution microscopes devised so far are structured illumination microscopy (SIM) [164], stimulated emission depletion microscopy (STED) [165], photo-activated localization microscopy (PALM) [166], and stochastic optical reconstruction microscopy (STORM) [167]. Utilization of these methods in the field of mitochondrial research significantly expanded over the last decade. On the one hand, three-dimensional SIM (3D SIM) imaging of mitochondria, labeled with MitoTracker Green, has resolved cristae structures, which were previously visible only in electron microscopy images [168]. On the other hand, STORM allowed for the visualization of the mitochondrial inner membrane dynamics by using MitoTracker probes in living cells [169]. Moreover, STED microscopy revealed an array of Mitochondrial INner membrane Organizing System (MINOS) clusters in human mitochondria, by showing the distribution of three individual MINOS subunits in mammalian cells [170]. STED has also been employed to demonstrate distinct mitochondrial ribosome clusters that interact with proteins involved in mRNA metabolism and respiratory chain assembly [171]. When it comes to the structural arrangement of mtDNA in nucleoids, 2D and 3D applications of PALM have enabled the visualization of ellipsoidal nucleoids containing extremely condensed mtDNA. The properties of the photoconvertible protein mEOS2 fused to the mitochondrial transcription factor TFAM allowed individual nucleoids to be resolved, to calculate their dimensions, and their distribution in the matrix [172].

Despite the clear gain in resolution reached with these techniques, it must be kept in mind that super-resolution microscopy is heavily time-consuming, and the available image analysis solutions are still not user-friendly. Moreover, the biggest challenge nowadays is to optimize these systems to perform dynamic live cell imaging while achieving super resolution. To this end, a recently developed imaging technique called MINFLUX combines STED and PALM/STORM approaches. Therefore, MINFLUX offers an increased resolution—by a factor of 100—at the nanometer scale, while imaging in living cells [173]. Another exciting alternative that can also allow for live cell imaging on a molecular scale is the lattice light sheet microscopy [174]. This technique has recently been used to demonstrate mitochondrial positioning according to the ATP:ADP gradient in mouse embryonic fibroblasts [175]. Technological and methodological upgrades, both in super-resolution approaches and lattice light sheet microscopy techniques, will certainly become the next frontier for the analysis of mitochondrial dynamics with an unprecedented spatial resolution in living cells.

## 4. Mitophagy

During stress conditions, mitochondria were shown to produce excessive amounts of ROS that can disrupt the organelle function in terms of ATP synthesis [176,177]. Increased levels of mitochondrial ROS can promote excessive Ca^2+^ uptake, leading to a loss in membrane potential and subsequently causing the release of apoptotic factors that can initiate cell death signalling [178,179,180]. In this context, mitochondrial quality control, and thus the maintenance of a healthy and functional mitochondrial population, is an important determinant of cell fate [181]. To this end, a selective form of autophagy—termed mitophagy—serves as a protective mechanism to selectively sequester and degrade damaged mitochondrial units before they activate cell death pathways [182,183]. 

Ever since the first observation of mitochondria inside an autophagosome in 1957, a considerable amount of work has been done to delineate the molecular pathways leading to mitophagy [184]. However, the field is still in need of robust live imaging tools to monitor and quantify mitophagy events in real time.

### 4.1. Molecular Pathways of Mitophagy

The most well studied molecular pathway of mitophagy is mediated by the phosphatase and tension homologue (PTEN)-induced putative kinase 1 (PINK1) and the E3 ligase PARK2 (Parkin) (Figure 1) [185]. Under basal conditions, PINK1 is imported into the IMM and cleaved by several intramitochondrial proteases [186,187,188,189]. Truncated PINK1 is then released into the cytoplasm and degraded by the ubiquitin-proteasome system [190,191]. Upon membrane potential loss, PINK1 can no longer be imported into the IMM, but it is rather stabilized on the OMM [192,193,194] where it recruits Parkin [192]. The E3 ligase activity of Parkin is activated when PINK1 phosphorylates ubiquitin at Ser 65 [195]. Parkin then ubiquitinates OMM proteins for their recognition and engulfment by the autophagosomes or degradation by proteasomal machinery [196,197,198,199,200,201]. In the case of autophagosomal engulfment, autophagy receptor proteins such as sequestosome 1 (SQSTM1)/p62 bind to the ubiquitinated mitochondrial units to link them with autophagosomes via microtubule-associated protein 1 light chain 3 (MAP1LC3, mostly known as LC3) [202]. Subsequent fusion of the autophagosomes with the lysosomes results in the complete degradation of mitochondrial remnants. This mechanism was shown to be impaired in inherited forms of Parkinson disease, in which the accumulation of dysfunctional mitochondria contributes to the death of dopaminergic neurons [203]. 

Aside from PINK1/Parkin-dependent mitophagy, mitochondrial clearance pathways regulated by other ubiquitin E3 ligases, such as Gp78, SMURF1, SIAH1, MUL1, and ARIH1, have been reported [204]. Similar to Parkin, these proteins can generate ubiquitin chains and modulate the recruitment of autophagy adaptor proteins [204]. A common trait between PINK1/Parkin-dependent and independent pathways is that several mitochondrial proteins appear to serve as mitophagy receptors. Due to their direct binding to LC3, they target dysfunctional mitochondria to autophagosomes for degradation [205]. A recent example of such a receptor is Prohibitin 2 (PHB2), which is an IMM protein shown to directly interact with LC3 after OMM rupture and to be indispensable for Parkin-dependent and independent mitophagy [206].

### 4.2. Roles of Mitophagy in Cellular Homeostasis

Under physiological conditions, mitophagy can occur at a basal rate to guarantee the normal turnover of metabolically inefficient organelles, or damaged parts of the mitochondrial network [207,208]. This rate is regulated in a tissue-specific manner to meet the varying metabolic needs of different tissues [209,210]. Additionally, basal mitophagy rates can be upregulated upon environmental stress factors such as starvation, hypoxia, and the administration of mitochondrial uncouplers to facilitate mitochondrial quality control [211,212,213]. Moreover, cells may perform programmed mitophagy events to enforce mitochondrial clearance during development [214,215], the degradation of paternal mitochondria upon fertilization [216], or when a metabolic switch from glycolysis to oxidative phosphorylation occurs [217,218]. A growing body of evidence shows that the complex interplay between these different mitophagy paradigms requires the utilization of different molecular players, and the crosstalk with mitochondrial dynamics and endosomal pathways [219,220,221].

### 4.3. Probes to Monitor Mitophagy

The first image-based strategy to observe mitophagy was performed with electron microscopy, where the mitochondria were engulfed by lysosomes and appeared to be in stages of “breakdown” or “hydrolysis” [222]. These early studies provided the first ultrastructure of mitophagy events, and although showing an unprecedented spatial resolution of this process, electron microscopy approaches have little temporal resolution. In light of this, strategies to image mitophagy have evolved to employ confocal or widefield fluorescence microscopy, as these image capabilities have a convenient spatiotemporal resolution and can be used in a quantitative mode. It should be noted that these strategies are often coupled with biochemical approaches such as Western blotting, to follow the degradation of specific mitochondrial markers. 

In the following section, we will focus on some of the commonly used probes that can be used to monitor mitophagy by fluorescence microscopy (Figure 1).

#### 4.3.1. Colocalization of Mitochondrial Probes with Autophagic or Lysosomal Markers

Colocalization analyses of mitochondrial fluorescent probes with markers of autophagosomes or lysosomes were one of the initial approaches to quantify mitochondrial units undergoing mitophagy. In these studies, LC3 was often used as a marker protein of autophagosomes, as it is tethered into the autophagic membranes upon the induction of autophagy [223]. LC3 can be monitored by fusing it with a fluorescent protein such as GFP. Mitochondria, however, can be labeled with a mitochondria-targeted red fluorescent protein (RFP) [112], or with MitoTracker dyes spectrally-compatible with GFP-LC3 [224]. Visualization and quantification of GFP-LC3 punctate structures colocalizing with mitochondrial markers can then be indicative of mitochondria undergoing mitophagy [225]. While this method has been used in many studies, it has two main limitations. First, LC3 was shown to accumulate into intracellular protein aggregates, which are independent of autophagy, thus making it difficult to separate random aggregation events from true mitophagy [223]. Second, the GFP fluorophore is quenched in environments with an acidic pH. Such drops in pH occur in the late stages of the mitophagy pathway, when the autophagosome fuses with the lysosome [225]. Therefore, this strategy cannot be used as a direct read-out of mitochondrial degradation, since the hydrolysis of mitochondria cannot be observed [226]. 

In order to monitor the degradation of mitochondria into lysosomes, lysosomal marker proteins such as the lysosomal-associated membrane protein 1 (LAMP1) can be fused with fluorescent proteins resistant to acidic pH (i.e. mCherry, DsRed2, and their derivatives). Fluorescent LAMP1 can then be used in combination with mitochondrial probes to follow the later events of mitophagy [227]. Alternatively, LysoTracker dyes can also be used to follow the colocalization of mitochondria with lysosomes [224,228,229]. These dyes have acidotropic properties, and therefore, they are recruited to the acidic compartments in the cell [230]. However, they have been reported to photobleach rapidly, thus posing a potential threat for long time-course experiments [231].

#### 4.3.2. Genetically-Encoded Mitophagy Sensors

Recently, several genetically-encoded mitophagy sensors have been developed to provide spatio-temporal information while monitoring mitophagy events in live cells. Considering the mitophagy flow as a dynamic course of events, these sensors have the potential to give detailed insights about the mitophagy pathway by sensing certain elements of this flux.

A tandem RFP-GFP fusion protein targeted to mitochondria is an example of genetically-encoded mitophagy sensor. It is designed to distinguish key steps in the mitophagy flux as the events of autophagosome formation and their fusion with lysosomes [232]. The idea behind this probe relies on the differences in the pK_a_ of GFP and RFP (c.f. previous chapter). When the autophagosomes are formed but not yet fused with the lysosomes, they have colocalized red and green signals of the punctate structures, which would indicate that the mitochondria are engulfed by the autophagosomes. Conversely, the red-only punctate structures can be used to identify autophagosomes fused with lysosomes. This is due to the acidic environment of lysosomes, which would cause the quenching of GFP, but not of RFP [232]. For instance, an OMM-targeted RFP-GFP tandem has been used to identify iron-chelators as PINK1/Parkin-independent mitophagy inducers [233], whereas an IMM-targeted RFP-GFP tandem was used to show mitophagy activation upon hepatitis B virus infection [234]. 

By following a similar experimental strategy, a transgenic mouse model with a pH-sensitive mitochondrial fluorescent probe was developed [207]. This construct was named mito-Quality Control (mito-QC), and it is constituted of a mCherry-GFP tandem protein targeted to the OMM via the MTS of the mitochondrial fission protein FIS1 [207]. By following the mCherry-only punctate structures, McWilliams et al. used mito-QC to monitor mitochondria targeted at lysosomes upon mitophagy activation.

Mt-Keima is another genetically-encoded fluorescent reporter used to follow mitophagy events in cells and in vivo [208,235]. Mt-Keima is a probe targeted to the mitochondrial matrix that can change its spectral properties depending on the surrounding pH [235]. When mt-Keima is in an environment with elevated or physiological pH, as in mitochondria (pH ~8.0), it has a green fluorescence. On the contrary, when pH drops as in lysosomes (pH ~4.5), mt-Keima becomes ionized and it shows a red fluorescence [235]. Mt-Keima has been used to follow the delivery of mitochondria to lysosomes upon mitophagy activation, and it was reported to be stable in lysosomes [235,236]. It should also be acknowledged that mt-Keima was shown to have limitations in terms of spectral separation between the green and the red forms [207], which therefore questions whether its use is pertinent for mitophagy. 

In addition to mitophagy, mitochondrial biogenesis and turnover can also be monitored to asses mitochondrial “aging” in cells. For this purpose, a probe named MitoTimer was developed by targeting the Timer fluorescent protein to the mitochondrial matrix [237,238]. Along the process of maturation, the Timer protein undergoes an oxidation, which induces an irreversible fluorescence shift from green to red over time [238]. Given that it works as a molecular clock, this probe was used to obtain spatiotemporal information on mitochondrial biogenesis (green organelles) and turnover (red mitochondria).

## 5. Conclusions

As described in this review, mitochondria are organelles with diverse functions and morphological features. While the complex nature of these organelles is important to regulate cellular homeostasis and metabolism, it also makes it challenging to follow and image them. However, the majority of studies have benefited from imaging probes targeted at mitochondria. In this context, our chance to deepen our knowledge about mitochondria and their function in health and disease relies on the availability of the microscopy-based tools, and recent advances in live imaging techniques. 

Benefiting from a variety of chemically and genetically-engineered reporters, mitochondrial Ca^2+^ buffering and signalling is the most extensively-studied mitochondrial function. However, the available probes for monitoring mitochondrial dynamics and mitophagy pathways are limited in terms of their ability to follow the complete flow of events. This is especially noteworthy for monitoring mitophagy events, as the majority of studies focus on the colocalization of LC3 with a mitochondria-targeted fluorescent probe. As discussed above, this can be misleading due to the lack of knowledge about the early and late time-points of the mitophagy pathway. In light of this, the investigation of alternate and specialized marker proteins recruited to autophagosomes will certainly pave the way to the design of new probes. 

Genetically-encoded sensors are advantageous compared to chemical reporters, as they are engineered to be substrate-specific, therefore allowing for the following of a detailed series of events over a long period of time [77,239]. An exciting challenge in this field would be the use of two—or potentially more—genetically-encoded sensors at the same time in order to be able to follow different mitochondrial functions simultaneously. This strategy could be further extended by using these biosensors in combination with chemical reporters, and also in screening modes. This would allow for the performing of high-content analyses of mitochondria in response to pharmaceutical compounds, or to discover new mitochondria-based therapeutic agents. 

Ongoing research to improve the spectral properties of the fluorescent proteins, and the advances in imaging methodologies, will certainly allow researchers to develop mitochondria-based diagnostic and drug-screening technologies in the near future. These technologies will eventually lay the groundwork for personalized medicine for people suffering from a mitochondrial disease, or from multifaceted pathologies where mitochondria play a key role.

## Figures and Tables

**Figure 1 genes-11-00125-f001:**
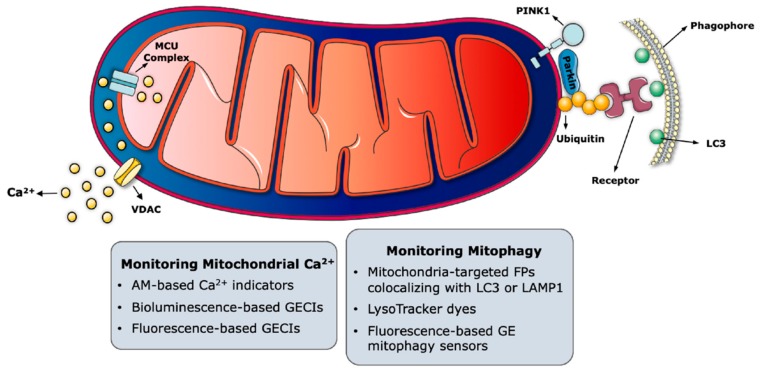
Probes to monitor mitochondrial Ca^2+^ signalling and mitophagy. The major molecular players regulating these functions are symbolized within mitochondria. Key imaging-based approaches are listed in the grey boxes in proximity to the corresponding mitochondrial function. MCU complex, mitochondrial Ca^2+^ uniporter complex; VDAC, voltage-dependent anion-selective channel proteins; PINK1, phosphatase and tension homologue (PTEN)-induced putative kinase 1; Parkin, E3 ligase PARK2; LC3, microtubule-associated protein 1 light chain 3; LAMP1, the lysosomal-associated membrane protein 1; AM, acetoxymethyl ester; GECIs, genetically-encoded Ca^2+^ indicators; FPs, fluorescent proteins; GE, genetically-encoded. The figure was generated with elements provided in the Servier Medical Art depository (https://smart.servier.com/) and are licensed under a Creative Commons Attribution 3.0 Unported License (CC BY 3.0).

**Figure 2 genes-11-00125-f002:**
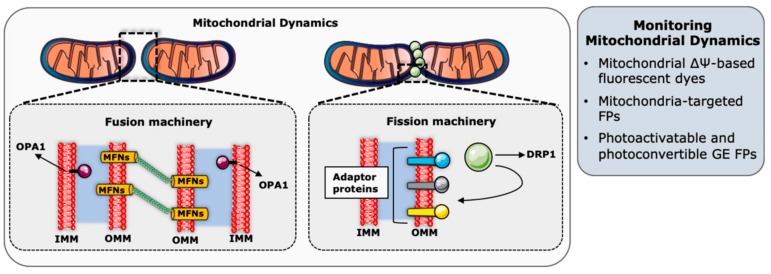
Probes to monitor mitochondrial dynamics. The major molecular players regulating these functions are symbolized within mitochondria. Key imaging-based approaches are listed in the grey box. OPA1, dynamin-like GTPase optic atrophy 1; MFNs, mitofusin 1 and mitofusin 2; DRP1, dynamin-related protein 1; IMM, inner mitochondrial membrane; OMM, outer mitochondrial membrane; ΔΨ, membrane potential difference; FPs, fluorescent proteins; and GE, genetically-encoded. The figure was generated with elements provided in the Servier Medical Art depository (https://smart.servier.com/) and are licensed under a Creative Commons Attribution 3.0 Unported License (CC BY 3.0).

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
