# Peer review of "Imaging Mitochondrial Functions: From Fluorescent Dyes to Genetically-Encoded Sensors"

_genes, 2020, doi:10.3390/genes11020125_

Round 1

Reviewer 1 Report

In the manuscript by Gökerküçük et al., the authors reviewed recent technological advances in imaging mitochondria. They focused on three aspects of mitochondrial biology: calcium homeostasis, mitochondrial dynamics and mitophagy. This manuscript is overall well-written and have included most of the up-to-date literatures. The content is suitable for the scope of the journal and the special issue. The manuscript will benefit by addressing the following comments:

1. The authors should discuss recent advances in imaging mitochondrial RNAs and mitochondrial RNA granules.

2. The authors should discuss studies using FRAP and FLIP to study diffusion properties within mitochondria.

3. The manuscript contains a number of grammar errors. For example, in line 1 & 16, “as” should be “such as”. Line 35, “to be able” can be deleted. Line 339, “researches” should be “researchers”. Please similar such errors in the manuscript.

4. In line 39, mtDNA should encode 13 protein subunits (others are tRNAs and rRNAs).

Author Response

ANSWERS TO REVIEWERS

Reviewer 1

In the manuscript by Gökerküçük et al., the authors reviewed recent technological advances in imaging mitochondria. They focused on three aspects of mitochondrial biology: calcium homeostasis, mitochondrial dynamics and mitophagy. This manuscript is overall well-written and have included most of the up-to-date literatures.

We thank Reviewer n°1 for taking the time to review our manuscript, and for his/her positive comments.

The content is suitable for the scope of the journal and the special issue. The manuscript will benefit by addressing the following comments:

The authors should discuss recent advances in imaging mitochondrial RNAs and mitochondrial RNA granules.

We agree with the reviewer that mitochondrial RNAs and RNA granules are important for the physiology of mitochondria. In this light, STED microscopy was shown to be particularly helpful in determining the interaction of RNA granules with the OXPHOS chain with unprecedented spatial resolution. We added this point in lines 400-401 of the present version of the manuscript.

The authors should discuss studies using FRAP and FLIP to study diffusion properties within mitochondria.

We thank the reviewer for this suggestion. We added an introductory sentence on FRAP and FLIP (lines 348-351) and we discussed about the importance of these techniques to study mitochondrial functions in lines 376-385.

The manuscript contains a number of grammar errors. For example, in line 1 & 16, “as” should be “such as”. Line 35, “to be able” can be deleted. Line 339, “researches” should be “researchers”. Please similar such errors in the manuscript.

We thank the reviewer for helping us to ameliorate the readability of our manuscript. We edited the text following his/her comments.

In line 39, mtDNA should encode 13 protein subunits (others are tRNAs and rRNAs).

Line 40 was edited according to this comment.

We hope that these modifications and additions will be considered sufficient for publication.

Reviewer 2 Report

The authors prepared a very well structured and sound review on how mitochondria can be imaged with the use of fluorescence microscopy in order to probe mitochondrial structure and function.

I have a few minor remarks concerning the presented techniques/probes which inclusion could be interesting for a general readership:

Chemical engineered Ca2+-probes (line 140-159): the author might want to include the fact, that some probes are also useful for quantitative Ca2+-imaging. Some could even be used to monitor changes of Ca2+-levels without calibration – the ratiometric dye Fura-2 would be an example. Organic probes vs. GECIs (lines 140-226): the authors mention several advantages of GECIs, however, as the authors are aware of, their large size compared to an organic probe or just the abundant expression of proteins could influence mitochondrial functions. - The membrane potential of mitochondria (lines 308-322) is the driving force for live-sustainable processes. I wonder why the authors hide this important property of mitochondria in the section “Mitochondrial Dynamics”?
- Nowadays there are more advanced version of MitoTracker green available – the authors should mention the most important ones (e.g. rosamine-based,…). Super-resolution microscopy for mitochondrial ultrastructure (lines364-387): the authors only mention, that some of these techniques are heavily time-consuming. The more important consequence arising from this fact is, that they are hardly applicable in live cell (mitochondria) imaging and most (if not all) mentioned examples represent experiments performed on chemically fixed samples. The authors should also mention lattice light sheet microscopy, which is capable of imaging live cell dynamics – even on a molecular scale (e.g. the original paper https://www.ncbi.nlm.nih.gov/pubmed/25342811 and more recent results). MINFLUX (https://www.ncbi.nlm.nih.gov/pubmed/28008086 and https://www.ncbi.nlm.nih.gov/pubmed/31819263) might be also a super-resolution way to go for live imaging.

Author Response

Reviewer 2

The authors prepared a very well structured and sound review on how mitochondria can be imaged with the use of fluorescence microscopy in order to probe mitochondrial structure and function.

We would like to thank Reviewer n°2 for taking the time to review our manuscript, for his/her appreciation of our work and for the insightful comments provided.

I have a few minor remarks concerning the presented techniques/probes which inclusion could be interesting for a general readership:

Chemical engineered Ca2+-probes (line 140-159): the author might want to include the fact, that some probes are also useful for quantitative Ca2+-imaging. Some could even be used to monitor changes of Ca2+-levels without calibration – the ratiometric dye Fura-2 would be an example. Organic probes vs. GECIs (lines 140-226): the authors mention several advantages of GECIs, however, as the authors are aware of, their large size compared to an organic probe or just the abundant expression of proteins could influence mitochondrial functions.

Following the reviewer’s considerations, we now indicate that probes as Fura-2 and Rhod-2 can have a quantitative readout in determining the exact Ca2+ concentration in mitochondria (Lines 163-165). In addition, we indicated that GECIs were also used to this end (lines 179-180).

Concerning GECIs, we now mention that their big size compared to organic probes and/or expression levels could compromise mitochondrial functions (lines 177-179). We thank the reviewer for these critical comments, as these drawbacks of GECIs are important for newcomers in the field.

 - The membrane potential of mitochondria (lines 308-322) is the driving force for live-sustainable processes. I wonder why the authors hide this important property of mitochondria in the section “Mitochondrial Dynamics”?

We agree with the reviewer on the importance of mitochondrial membrane potential for mitochondrial and cell physiology. Indeed, we talk about the fundamental role of DY in the “Calcium signaling” (Paragraph 2.1 and Figure 1) and in the “Mitochondrial dynamics” sections (Paragraph 3.2.1)

- Nowadays there are more advanced version of MitoTracker green available – the authors should mention the most important ones (e.g. rosamine-based,…).

In the present version of the manuscript, we now provide an additional sentence and new references concerning the other MitoTracker dyes available (Lines 326-328). We also discuss the sensitivity of rosamine-based MitoTrackers to DY, which is still a matter of debate in the field and would be interesting to the general readership.

- Super-resolution microscopy for mitochondrial ultrastructure (lines364-387): the authors only mention, that some of these techniques are heavily time-consuming. The more important consequence arising from this fact is, that they are hardly applicable in live cell (mitochondria) imaging and most (if not all) mentioned examples represent experiments performed on chemically fixed samples. The authors should also mention lattice light sheet microscopy, which is capable of imaging live cell dynamics – even on a molecular scale (e.g. the original paper https://www.ncbi.nlm.nih.gov/pubmed/25342811 and more recent results). MINFLUX (https://www.ncbi.nlm.nih.gov/pubmed/28008086 and https://www.ncbi.nlm.nih.gov/pubmed/31819263) might be also a super-resolution way to go for live imaging.

We thank the reviewer for these helpful suggestions, which allow us to further underline the importance of super-resolution microscopy for the field of mitochondria in the years to come. We are aware that the next frontier in mitochondrial imaging will be to combine super-resolution microscopy and live cell imaging. We now integrated our paragraph on super-resolution microscopy (lines 408-418) by describing the recent advances brought by the MINFLUX approach and lattice light sheet microscopy for live mitochondrial imaging, and integrating the corresponding references as suggested. Concerning the last reference suggested by the reviewer, we felt that the evidence linking MINFLUX and live cell imaging was fragmentary in the corresponding paper. In this light, we preferred not to cite it. 

We hope that the reviewer will be satisfied with our modifications, and that our manuscript will be judged suitable for publication.